# The impact of Evidence-Based Pharmacy on the quality of pharmaceutical care: A survey study

Piotr Ratajczak[1]*, Barbara Jacieczko[1], Bartosz Kędziora[2], Dorota Kopciuch[1], Anna Paczkowska[1], Tomasz Zaprutko[1], Krzysztof Kus[1]

**1** Department of Pharmacoeconomics and Social Pharmacy, Poznan University of Medical Sciences, Poznań, Poland, **2** Collegium Medicum, Jan Kochanowski University, Kielce, Poland

\* p_ratajczak@ump.edu.pl

## Abstract

Applying Evidence-Based Pharmacy (EBPharm) principles is essential in pharmacy practice as it allows therapeutic decisions to be grounded in the best available scientific evidence. This approach enables pharmacists to provide patients with more effective and safer therapies, reducing the risk of adverse effects. This study assessed the EBPharm knowledge level among pharmacy professionals. Data were collected using a custom-designed online survey. The study also explored and defined the concept of EBPharm, offering insights into medical databases, types of scientific research, and evaluations of their quality and reliability. The study revealed widespread application of EBPharm principles among Polish pharmacy professionals, with significant associations between their use and factors such as professional experience (Chi² = 7.905518, p = 0.04801) and higher salaries. Respondents with formal EBPharm training demonstrated improved ability to evaluate scientific evidence (Chi² = 12.03125, p = 0.00244), while financial stability was linked to better systematic review skills (Chi² = 43.15693, p = 0.00000) and use of databases (Chi² = 63.14029, p = 0.00000). However, knowledge gaps remain, particularly among less experienced professionals, with only 30.35% of participants receiving formal training and 8.42% deeming it sufficient. Frequent use of informal sources by pharmacy technicians and limited familiarity with frameworks like PICO (Chi² = 37.65817, p = 0.00000) highlight the need for accessible, structured education. Targeted training, financial incentives, and greater reliance on reliable sources could enhance EBPharm integration, improving care quality and patient safety.

## Introduction

Evidence-Based Pharmacy (EBPharm, EBP) stems from the concept of Evidence-Based Medicine (EBM). It aims to use the best current scientific evidence from research to inform decision-making in pharmaceutical practice. EBPharm focuses on making therapeutic decisions based on scientific data from well-designed and conducted studies.

**Data availability statement:** All relevant data are within the manuscript and its Supporting information files.

**Funding:** The author(s) received no specific funding for this work.

**Competing interests:** The authors have declared that no competing interests exist.

Given the rapid growth of medical information and the increasing number of available drugs, pharmacists must thoroughly analyse and interpret research to provide patients with medications suited to their needs and financial capacity while offering high-quality consultation. EBPharm primarily pertains to pharmacological treatment, broadly understood pharmacotherapy, drug use, resolving drug-related problems (such as interactions), pharmaceutical care, and education. Its goal is to improve the effectiveness and efficiency of both short- and long-term treatments while also enhancing the quality of pharmacists' work.

Pharmacists often act as the first, and sometimes only, healthcare system consultants when it comes to questions related to self-medication. The rational use of pharmacists' skills and advice in self-care can contribute to the provision of cost-effective, personalised, and safe healthcare at the patient level. In turn, this can help alleviate the burden on the overall healthcare system. A broad knowledge base is essential for consultations and is a crucial element within EBPharm [1].

Pharmacists using the latest evidence can better manage patient therapies, improving health outcomes. Research shows that evidence-based pharmaceutical interventions can significantly improve the control of chronic diseases like diabetes and hypertension through more precise and effective treatments [2]. Enhancing patient safety is another advantage of EBPharm, as applying the latest research helps identify and minimise the risks associated with drug therapy. Pharmacists who regularly update their knowledge are better equipped to detect potential drug interactions and counteract adverse effects [3].

Cost optimisation in healthcare is also an essential aspect of EBPharm, as it allows pharmacists to select the most cost-effective therapies, benefiting both patients and the healthcare system. Studies indicate that proper drug management can lead to substantial savings by reducing hospitalisations and decreasing the need for additional medical interventions [4]. Another benefit of regularly using the latest scientific research is raising the standards of pharmaceutical care. It enables pharmacists to deliver services at the highest level, which helps boost professionalism and trust among patients [4].

EBPharm also fosters interdisciplinary collaboration, promoting better cooperation between pharmacists and other healthcare professionals, leading to more coordinated and comprehensive patient care ultimately improving the overall quality of care [5]. Another significant element of EBPharm is enhancing pharmacists' job satisfaction. Pharmacists with access to the latest scientific evidence and know how to apply it effectively experience greater job satisfaction. Knowledge and the ability to use it in clinical practice lead to greater confidence and fulfilment in their work [6].

Pharmacist education in effectively searching for and evaluating scientific evidence is crucial to avoid access to conflicting information. It is more important to learn how to find reliable sources than to focus solely on the speed of searching, as low-quality information can significantly impact the quality of consultations provided. Therefore, training in assessing the quality of scientific evidence is also essential [7].

Education should foster critical thinking, problem-solving, and decision-making skills in pharmacotherapy. Pharmacists should be trained during their studies to

create, share, and use knowledge based on the latest research and collaborate with other healthcare professionals to improve patients' quality of life by improving their health [8].

This study provides valuable insights into applying EBPharm principles among Polish pharmacy professionals, highlighting achievements and improvement areas. Additionally, the study explored the respondents' preferences regarding sources of scientific information and analysed their awareness of the quality and reliability of scientific publications available in medical databases.

## Materials and methods

### Study design

The cross-sectional survey was selected for this study as it enables efficient data gathering from a specific group of individuals. This approach is appropriate for capturing and representing various respondents' experiences and behaviours. Results reporting was based on the checklist for survey studies [9].

The questionnaire was developed based on a literature review in EBPharm, pharmacy education, and information literacy. Three key domains were identified: (1) knowledge and attitudes toward EBPharm, (2) use of scientific sources, and (3) the ability to evaluate the quality of scientific publications.

The creation of the tool was based on expert consultation and pilot testing to ensure thematic balance and respondent feasibility. To address content and face validity, all items were reviewed by academic experts in pharmacy and education. The questionnaire was internally tested within our academic unit before distribution.

### Data collection and research method

The inclusion criteria of an anonymous survey were working as a pharmacist or pharmaceutical technician in Poland's professional group employed in open (community pharmacies) or closed pharmacies (pharmacies operating within institutions such as hospitals) or being a pharmacy student currently undergoing a professional internship.

The survey contained 26 primary questions and six supplementary ones. The survey questions included demographic data of the respondents (such as gender, age, place of residence and population size, level of education, employment, income level, years of work experience, and specialisation in pharmacy) and questions related directly to the EBPharm topic. The term "pharmacy specialisation" refers to formal postgraduate specialisation training in Poland (e.g., in hospital pharmacy, clinical pharmacy, pharmacology, pharmacoeconomics). The questions covered topics related to knowledge and attitudes toward EBPharm (questions 10–14), the use of scientific sources (questions 15–22), and the ability to evaluate the quality of scientific publications (questions 23–25). Answer scales were adapted to question types: binary for factual items, Likert-type for self-assessment, and three-point scales for frequency-based questions.

The survey was conducted in Polish, with the questionnaire and results translated into English after completing the study.

### Informed consent statement

Written informed consent was obtained directly before participants proceeded with the survey. Participation in the survey was entirely voluntary, and respondents were informed about the purpose of the study, ensuring they understood their rights, including the option to withdraw at any time. Each participant confirmed their consent to data collection and processing by completing the survey.

### Sample characteristics

The minimal sample size of 266 participants for the professional group of pharmacists and pharmaceutical technicians in Poland was provided by Statistics Poland [10] which amounts to 57.8 thousand employees. The sample size was

calculated using Raosoft Calculator (www.raosoft.com) with a 6% margin of error (allows for a reasonable level of result accuracy while significantly reducing the required sample size, making the study more feasible), 95% confidence level, 57 800 population size and 50% response distribution.

### Survey administration

The survey was conducted from January 2, 2024, to April 14, 2024. A total of 313 surveys were collected and included in the analysis. The survey questionnaires were distributed via a generated link using 'Google Forms' directly in stationary pharmacies across several cities in Poland and by social media platforms to increase the reach (Facebook, Instagram and LindedIn). To ensure the respondents' identity, the survey's introduction included the sentence, "The survey is aimed at individuals working in pharmacies and at pharmacy students currently undergoing professional internships". Additionally, qualifying questions related to the level of education, job position, salary, and work experience were introduced. The survey was also published in specific thematic groups. All results that were obtained met the expectations regarding the target group.

### Ethics statement

The Poznan University of Medical Sciences Bioethics Committee approved the study (December 6, 2023). The study did not have the characteristics of an experiment.

### Statistical analysis

The results were subjected to descriptive analysis (single-choice and multiple-choice questions, ranking and matrix questions) and statistical analysis using Pearson's $Chi^2$ independence test (single-choice questions). The study was conducted using Statistica 13.3 software (StatSoft, TIBCO). Pearson's $Chi^2$ coefficient values with $p < 0.05$ were considered statistically significant. For the analysis of several multiple-choice question data (5, 17, 20.2, 21.1.), each possible answer was converted into a binary variable (yes/no) and then checked for correlations with other responses (Supplement – Survey Data file). Additionally, effect sizes were calculated using Cramer's V with 95% confidence intervals estimated via non-parametric bootstrapping (1,000 replicates), based on predefined variable pairs.

## Results

### Characteristics of the Study Group

The first nine survey questions were related to the basic information about the respondents. A total of 313 individuals participated in the study. Results related to the place of residence (question 4) were not reported based on no statistical correlations with other factors. Additionally, responses related to questions 8 (plus 8.1) and 9 were also not reported because of the low number of responses (respectively n = 6 and n = 16). The remaining results are presented in Table 1.

### Analysis of survey results

The remaining part of the survey aimed to assess opinions, preferences, and the level of knowledge regarding Evidence-Based Pharmacy (questions 10–14). It also focused on sources of scientific information and awareness of the quality and reliability of scientific publications available in medical databases (questions 15–25). The results are presented in Table 2 (single-choice questions), Table 3 (multiple-choice questions), and Table 4 (correlation between questions and statistical significance description). Additionally, correlations between questions are presented in S1 Table (supplementary file). The results for ranking question (20.3) and matrix question (24) are illustrated in Figs 1 and 2.

**Table 1. Characteristics of the study group.**

| Question | Answer | Result (%) |
|---|---|---|
| Gender (n = 313) | Female | 82.11 |
| | Male | 17.89 |
| Age (n = 313) | ≤26 | 11.18 |
| | 26-39 | 69.97 |
| | 40-59 | 15.97 |
| | ≥60 | 2.88 |
| Level of education (n = 313) | Student | 4.47 |
| | Pharmacy technician | 23.00 |
| | Higher education, first cycle (bachelor's, master's) | 71.25 |
| | Higher education, second cycle (doctorate) | 1.28 |
| Job position (n = 313) | Pharmacist in an open pharmacy | 71.25 |
| | Pharmacist in a closed pharmacy | 0.64 |
| | Pharmacy technician | 23.32 |
| | Pharmaceutical industry employee | 1.60 |
| | Research worker | 0.64 |
| | Currently unemployed | 5.11 |
| | Other (Pharmacist with private office, self-employed) | 0.32 |
| Average net monthly salary (n = 313) | ≤2500 PLN (≤ 580 €)* | 0.32 |
| | 2500-5000 PLN (580–1160 €)* | 33.55 |
| | 5000-7000 PLN (1160–1630 €)* | 48.24 |
| | ≥7000 PLN (≥ 1630 €)* | 11.82 |
| | Currently not employed | 6.07 |
| Years of professional experience (n = 313) | 0-5 years | 47.60 |
| | 6-10 years | 26.84 |
| | 11-19 years | 19.81 |
| | ≥20 years | 5.75 |
| Pharmacy specialisation (n = 313) | Yes | 1.92 |
| | No | 98.08 |

*at an average exchange rate of 1 EUR = 4.30 PLN (October 17th 2024).

Source: Own work based on survey data.

## Discussion

Evidence-based pharmacy is the foundation of modern pharmaceutical practice, focusing on applying the best available scientific evidence in daily pharmaceutical care. The primary aim of EBPharm is to optimise therapeutic outcomes and minimise the risk of adverse effects by employing scientifically proven treatment methods. In addition to pharmacists, policymakers utilise EBPharm principles to make informed therapeutic decisions [1,11]. EBPharm is also significant from an economic perspective, allowing decisions that benefit both the patient and the healthcare system [3,4].

Most respondents reported applying EBPharm principles in their practice, as evidenced by a statistically significant relationship between practice and experience levels. Interestingly, this result underscores a broad recognition of EBPharm's importance across varying levels of professional tenure. Higher salaries were also associated with an increased likelihood of applying these principles in practice, reflecting the potential influence of financial stability on professional engagement. A 2023 study on EBPharm among pharmacists working in community pharmacies [12] showed that although a significant number of pharmacists follow EBPharm principles, there is still room for improvement, particularly in using independent, peer-reviewed sources of information.

**Table 2. Summary of responses to single-choice questions.**

| Question number | Question | Answer | Result (%) |
|---|---|---|---|
| 10 (n=313) | Do you apply the principles of Evidence-Based Pharmacy (based on the concept of Evidence-Based Medicine) in your professional practice? | Yes | 98.21 |
| | | No | 4.79 |
| 11 (n=313) | How would you rate your knowledge of Evidence-Based Pharmacy? | Very good | 12.46 |
| | | Good | 41.85 |
| | | Average | 36.74 |
| | | Poor | 7.67 |
| | | I have no knowledge of it | 1.28 |
| 12 (n=313) | Were you trained in Evidence-Based Pharmacy during your studies or professional practice? | Yes | 30.35 |
| | | No | 69.65 |
| 12.1 (n=95) | If yes, was this training sufficient? | Yes | 8.42 |
| | | No | 91.58 |
| 13 (n=313) | Do you think Evidence-Based Pharmacy influences/could influence the quality of pharmaceutical care and increases patient safety? | Yes | 88.50 |
| | | No | 0.96 |
| | | I don't know | 10.54 |
| 14 (n=313) | Do you feel the need for further education in Evidence-Based Pharmacy? | Yes | 98.08 |
| | | No | 1.92 |
| 15 (n=313) | Do you check the reliability of the sources you read? | Yes, always | 49.84 |
| | | Yes, sometimes | 49.84 |
| | | No, never | 0.32 |
| 16 (n=313) | Do you regularly update your knowledge using the latest scientific publications? | Yes, always | 32.91 |
| | | Yes, sometimes | 66.13 |
| | | No, never | 0.96 |
| 18 (n=313) | Are you able to assess the level of scientific evidence in pharmacy-related research articles? | Yes | 60.38 |
| | | No | 38.66 |
| | | Other (I don't know) | 0.96 |
| 19 (n=313) | Do you know the PICO framework (Population, Intervention, Comparison, Outcome) – a tool for evaluating the quality of information? | Yes | 25.88 |
| | | No | 74.12 |
| 20 (n=313) | Do you use scientific evidence such as research publications in your work? | Yes, often | 22.04 |
| | | Yes, sometimes | 73.48 |
| | | No, never | 4.47 |
| 20.1 (n=299) | Is the affiliation (institutional association) of the authors of a scientific paper and funding information important for your analysis of the paper? | Yes | 95.32 |
| | | No | 4.68 |
| 21 (n=313) | How often do you use medical databases in your pharmaceutical practice or studies? | Daily | 0.96 |
| | | Several times a week | 5.43 |
| | | Several times a month | 38.66 |
| | | Less than once a month | 43.77 |
| | | I do not use them | 11.18 |
| 22 (n=313) | Are you able to conduct a systematic review of the scientific literature in the field of pharmacy? | Yes, entirely | 3.83 |
| | | Yes, partially | 46.65 |
| | | No | 49.52 |
| 23 (n=313) | Are you able to assess the quality and reliability of scientific research and pharmacy-related work published in scientific journals and available in medical databases? | Yes | 69.65 |
| | | No | 30.35 |
| 25 (n=313) | Are you able to analyse research results and translate them into understandable clinical information for patients or other healthcare professionals? | Definitely yes | 23.00 |
| | | Definitely yes | 73.81 |
| | | Probably no | 3.19 |

Source: Own work based on survey data.

Table 3. Summary of responses to multiple-choice questions.

| Question number | Question | Answer | Number of indications (%) |
|---|---|---|---|
| 17 (n=313) | Which of the following scientific information sources do you use in your work? | Publications in scientific journals | 76.04 |
| | | Conference reports | 61.34 |
| | | The Internet (forums, blogs, social media) | 60.06 |
| | | Academic books | 46.33 |
| | | Professional training/courses | 36.10 |
| | | Webinars | 33.55 |
| 20.2 (n=299) | What criteria do you consider when selecting a scientific publication? | Publication date | 93.98 |
| | | Journal reputation | 64.88 |
| | | Funding information | 58.86 |
| | | Accessibility of the publication in databases | 51.51 |
| | | Research methodology | 48.83 |
| | | Information on conflicts of interest | 48.83 |
| | | Presentation of results | 35.45 |
| | | Reputation of the institution | 22.74 |
| | | Author's name(s) | 5.35 |
| 21.1 (n=278) | Please indicate the medical databases you use. | PubMed | 94.96 |
| | | Google Scholar | 31.29 |
| | | Cochrane Library | 26.98 |
| | | Science Direct | 12.59 |
| | | Scopus | 10.07 |
| | | Web of Science | 8.99 |
| | | Embase | 7.19 |
| | | Other | 0.36 |

Source: Own work based on survey data.

Although most respondents rated their knowledge as good, the answers to detailed questions indicated gaps in their actual understanding and application of EBPharm principles in practice. An example is the irregular updating of knowledge based on the latest publications, the inability of a significant portion of respondents to assess the level of scientific evidence in scientific articles, unfamiliarity with the PICO framework, and the infrequent use of scientific evidence in their work. Additionally, with widespread application, self-assessed knowledge of EBPharm was inversely correlated with professional experience. This finding suggests a potential decline in confidence or knowledge retention over time, emphasising the need for continued professional development. Studies by Hale Toklu [1] and Christine Bond [13] also highlight the need for education in the EBPharm area. Respondents with higher salaries reported greater knowledge levels, highlighting financial incentives' potential role in fostering learning and skill development.

Formal EBPharm training significantly enhanced participants' ability to assess the quality of scientific evidence, indicating that targeted education can bridge critical knowledge gaps. However, the lack of proper training among those who hold this belief may result in an incorrect interpretation of the evidence, highlighting the need for further education in this area, which is also emphasised by other researchers [1,12]. Similarly, the likelihood of the ability to assess scientific evidence increased with higher earnings.

While 69.65% of respondents reported the ability to assess the reliability of scientific studies, the data revealed significant variability depending on training and professional background. Additionally, the likelihood of verifying the reliability of sources increased with salary. The results also suggest that while pharmacy professionals know the need to verify sources, this is not always a routine practice, as confirmed by other studies [14]. Older studies may be outdated and

**Table 4. Statistical analysis – selected results.**

| Question | | Question VERSUS question (section number/question number) | Correlation | Interpretation |
|---|---|---|---|---|
| Do you apply the principles of Evidence-Based Pharmacy (based on the concept of Evidence-Based Medicine) in your professional practice? | | 10 vs 7<br>Chi² = 7.905518<br>df = 3<br>p = 0.04801<br>CramersV = 0.170<br>(0.068–0.295) | Applying EBM principles **vs** work experience | Pharmacists and other pharmacy employees apply the principles of EBPharm in their professional practice, regardless of work experience. |
| How would you rate your knowledge of Evidence-Based Pharmacy? | | 11 vs 6<br>Chi² = 41.42779<br>df = 16<br>p = 0.00048<br>CramersV = 0.207<br>(0.163 - 0.260) | Level of knowledge **vs** salary level | The level of knowledge about EBPharm increases with salary level. |
| | | 11 vs 7<br>Chi² = 27.82321<br>df = 12<br>p = 0.00587<br>CramersV = 0.203<br>(0.141–0.268) | Level of knowledge **vs** work experience | The longer the professional experience, the lower the self-assessed knowledge about EBPharm. |
| Do you think Evidence-Based Pharmacy influences/could influence the quality of pharmaceutical care and increases patient safety? | | 13 vs 10<br>Chi² = 30.62768<br>df = 2<br>p = 0.00000<br>CramersV = 0.312<br>(0.122–1.491) | Quality of care and patient safety **vs** applying EBM principles | Those who apply EBPharm principles believe they improve the quality of care and patient safety |
| Do you feel the need for further education in Evidence-Based Pharmacy? | | 14 vs 5<br>Chi² = 25.62696<br>df = 9<br>p = 0.00235<br>CramersV = 0.329<br>(0.046–0.667) | Education in EBPharm **vs** job position | Regardless of job position, respondents feel the need for further education in EBPharm. |
| Do you check the reliability of the sources you read? | | 15 vs 6<br>Chi² = 41.61648<br>df = 8<br>p = 0.00000<br>CramersV = 0.266<br>(0.199–0.331) | Checking the reliability of sources **vs** salary level | The likelihood of checking the reliability of sources increases with salary level. |
| Which of the following scientific information sources do you use in your work? | Publications in scientific journals | 17 vs 6<br>Chi² = 19.18473<br>df = 4<br>p = 0.00072<br>CramersV = 0.271<br>(0.175–0.379) | Use of scientific journal publications **vs** salary level | The higher the salary, the more often respondents use scientific journal publications. |
| | The Internet (forums, blogs, social media) | 17 vs 4<br>Chi² = 22.52680<br>df = 3<br>p = 0.00005<br>CramersV = 0.231<br>(0.129–0.337) | Use of Internet **vs** education level | Pharmacy technicians more often use the Internet. |
| | Professional training/ courses | 17 vs 2<br>Chi² = 24.83544<br>df = 3<br>p = 0.00002<br>CramersV = 0.285<br>(0.198–0.380) | Use of professional training and courses **vs** age | Interest in professional training and courses increases with age. |

*(Continued)*

| Question | Question VERSUS question (section number/question number) | Correlation | Interpretation |
|---|---|---|---|
| | 17 vs 6<br>$Chi^2 = 15.26899$<br>df = 4<br>p = 0.00417<br>CramersV = 0.236<br>(0.132–0.334) | Use of professional training and courses **vs** salary level | The higher the salary, the less often training and courses are used. |
| | 17 vs 7<br>$Chi^2 = 27.89552$<br>df = 3<br>p = 0.00000<br>CramersV = 0.306<br>(0.206–0.402) | Use of professional training and courses **vs** work experience | The longer the professional experience, the greater the interest in training and courses. |
| Are you able to assess the level of scientific evidence in pharmacy-related research articles? | 18 vs 6<br>$Chi^2 = 32.90017$<br>df = 8<br>p = 0.00006<br>CramersV = 0.253<br>(0.173–0.343) | Assessment of evidence **vs** salary level | The ability to assess scientific evidence increases with salary level. |
| | 18 vs 12<br>$Chi^2 = 12.03125$<br>df = 2<br>p = 0.00244<br>CramersV = 0.203<br>(0.101–0.302) | Assessment of evidence **vs** EBPharm training | Those who have completed EBPharm training can assess the level of scientific evidence. |
| Do you know the PICO framework (Population, Intervention, Comparison, Outcome) – a tool for evaluating the quality of information? | 19 vs 4<br>$Chi^2 = 37.65817$<br>df = 3<br>p = 0.00000<br>CramersV = 0.257<br>(0.142 - 0.369) | Knowing PICO **vs** education level | Respondents, regardless of education level, do not know the PICO framework. |
| | 19 vs 6<br>$Chi^2 = 12.19712$<br>df = 4<br>p = 0.01594<br>CramersV = 0.216<br>(0.106 - 0.321) | Knowing PICO **vs** salary level | Knowledge of the PICO framework increases with salary level |
| Do you use scientific evidence such as research publications in your work? | 20 vs 4<br>$Chi^2 = 19.61690$<br>df = 6<br>p = 0.00324<br>CramersV = 0.136<br>(0.081–0.194) | Using scientific evidence **vs** education level | Respondents, regardless of education level, rarely use scientific evidence. |
| How often do you use medical databases in your pharmaceutical practice or studies? | 21 vs 6<br>$Chi^2 = 63.14029$<br>df = 16<br>p = 0.00000<br>CramersV = 0.254<br>(0.202–0.310) | Using medical databases **vs** salary level | The frequency of using databases increases with salary level. |
| | 21 vs 7<br>$Chi^2 = 29.67020$<br>df = 12<br>p = 0.00313<br>CramersV = 0.219<br>(0.150–0.309) | Using medical databases **vs** work experience | The frequency of using databases decreases with work experience. |

*(Continued)*

**Table 4.** (Continued)

| Question | Question VERSUS question (section number/question number) | Correlation | Interpretation |
|---|---|---|---|
| **Are you able to conduct a systematic review of the scientific literature in the field of pharmacy?** | 22 vs 6<br>$Chi^2 = 43.15693$<br>$df = 8$<br>$p = 0.00000$<br>CramersV = 0.278<br>(0.195–0.370) | Ability to assess the quality and reliability **vs** ability to conduct a systematic review | The ability to conduct systematic reviews increases with salary level. |
| **Are you able to assess the quality and reliability of scientific research and pharmacy-related work published in scientific journals and available in medical databases?** | 23 vs 12<br>$Chi^2 = 6.913873$<br>$df = 1$<br>$p = 0.00855$<br>CramersV = 0.142<br>(0.045–0.235) | Ability to assess the quality and reliability **vs** EBPharm training | Regardless of training (whether it was completed or not), respondents can assess the quality and reliability of data. |
| **Are you able to analyse research results and translate them into understandable clinical information for patients or other healthcare professionals?** | 25 vs 6<br>$Chi^2 = 33.10055$<br>$df = 8$<br>$p = 0.00006$<br>CramersV = 0.245<br>(0.186–0.309) | Ability to analyse and transform the results **vs** salary level | The ability to analyse research results and transform them increases with salary level. |
| | 25 vs 7<br>$Chi^2 = 15.39115$<br>$df = 6$<br>$p = 0.01742$<br>CramersV = 0.183<br>(0.107–0.262) | Ability to analyse and transform the results **vs** work experience | The ability to analyse research results and transform them increases with work experience. |

Source: Own work based on survey data.

inconsistent with new guidelines and standards of EBM (EBPharm), which can lead to obsolete or ineffective treatment methods [15]. Other authors [16] also indicate that up-to-date data is recommended and required in the educational process.

Another critical issue in the survey was the analysis of respondents' awareness of the quality and reliability of research and pharmacy work published in scientific journals and available in medical databases. Respondents trained in EBPharm were more adept at evaluating evidence quality. It was also observed that respondents with higher education and higher positions in a pharmacy are significantly more proficient in this area, which may be related to a different curriculum that lasts longer and requires mastering a more significant amount of information and knowledge in the case of pharmacists compared to technicians.

While many respondents reported partial ability to conduct systematic reviews (despite lacking formal training), nearly half lacked confidence in this skill. Only a small subset feels competent enough to perform such a review, which is surprising given that the vast majority of the respondents declared a high level of EBPharm application. Here, we observe the so-called 'iceberg effect', where people overestimate their knowledge and skills based on a superficial understanding of the topic. Many researchers overestimate their abilities in research methodology, leading to inaccurate and incomplete systematic reviews [17,18]. This inconsistency in the results becomes even more apparent when we consider that only a limited proportion of participants declared familiarity with the PICO framework, a fundamental tool used to formulate clinical questions and evaluate evidence. Knowledge of this framework is essential in the field of EBPharm, and such a contrast highlights a potential discrepancy between the perceived and actual implementation of expertise in this area. Although respondents may identify with the EBPharm concept, their understanding often appears superficial, based more on general awareness of the terminology than on genuine training. It may indicate the presence of self-reporting bias,

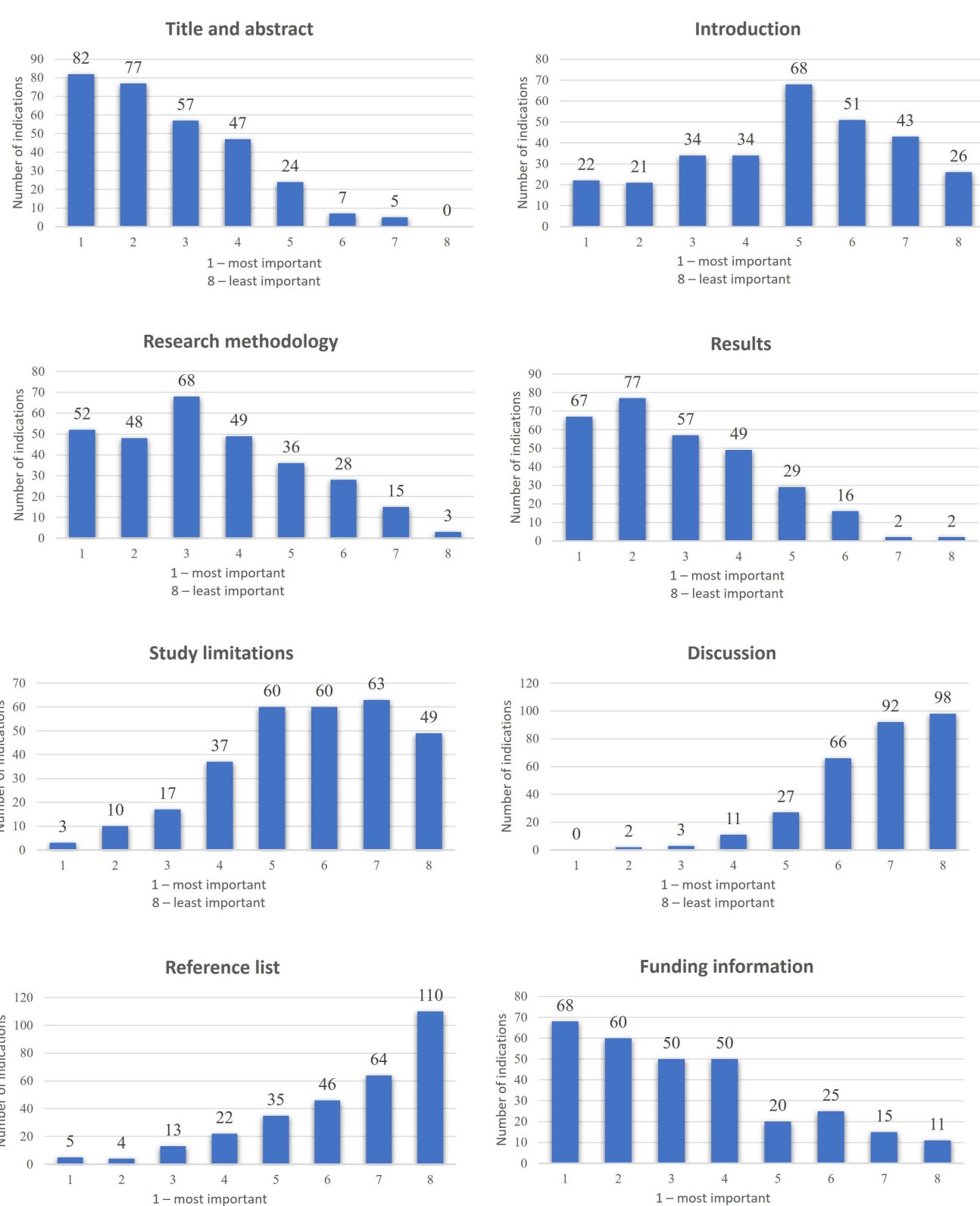

**Fig 1. Ranking publication sections (n = 299).** The figure presents the ranking of the importance of individual manuscript sections according to the respondents' opinions.

**How would you rate the reliability of the following types of research?**

**Fig 2. Study types reliability level (n = 313).** Respondents' opinions on the level of reliability of different types of scientific publications/studies.

where factors such as professional expectations lead individuals to overestimate their competencies. It points to the need to expand professional training for pharmacists.

Furthermore, higher income positively influenced the ability to conduct systematic reviews. Because of that, this area needs significant improvement as a key element of EBPharm, which enables the direct application of research findings in medical and pharmaceutical practice.

Interestingly, despite insufficient thematic training, most respondents can analyse research results and translate them into understandable clinical information for the patient. The ability to analyse research and translate it into clinical information was positively associated with work experience and salary increase. These results emphasise the interconnectedness of financial stability, education, and professional competence.

Respondents overwhelmingly agreed that EBPharm improves care quality and patient safety. It reinforces prior literature emphasising the role of evidence-based approaches in optimising patient outcomes through better-informed decision-making. Such an approach highlights the desire to deepen understanding, which could be essential to further developing evidence-based pharmaceutical practices [6].

The survey identified a strong demand for further education in EBPharm, regardless of job position. It aligns with the observation that only a minority of participants received formal training and among those, very few considered it sufficient. Interestingly, the frequency of attending professional training and courses decreased with higher salaries, suggesting that well-compensated professionals may rely on informal learning or feel more self-sufficient. Nevertheless, structured education remains crucial to addressing knowledge gaps.

A statistically significant relationship was observed between age and experience vs interest in professional training and courses, with older and more experienced respondents demonstrating a higher inclination towards formal education. Conversely, pharmacy technicians relied significantly more on informal internet sources, such as forums and social media, than pharmacists. This dichotomy highlights the need for structured, formal training accessible to all professional levels to reduce dependency on less reliable sources.

Despite recognising the value of scientific evidence, respondents across all education levels reported infrequent usage of scientific publications. It suggests that most respondents know the need to use reliable sources. However, this does not guarantee that these sources are always correctly interpreted [19]. This trend extends to medical databases, with usage

frequency inversely related to work experience. Medical databases, such as PubMed or the Cochrane Library, are invaluable tools in EBPharm practice [20,21]. However, the study results suggest that not all users are equally proficient in using them. Respondents often use medical databases less than once or several times a month. For many of them, the barrier may be access and the ability to search and filter information effectively. Indeed, English proficiency plays a role here as well, as it is an essential skill for researchers and those working in the scientific field, especially in the context of writing scientific publications and searching for information in medical databases. A study published in 2020 [22] demonstrates how much of an obstacle the lack of English proficiency can be and also highlights the issue of articles being rejected or requiring revision due to grammatical errors made by authors who are not fluent in English.

Higher salaries were associated with more frequent use of scientific journal publications and databases. PubMed emerged as the most frequently accessed database, underscoring its pivotal role in evidence-based practice. It may be due to its popularity, accessibility, and scope – it offers free access to over 37 million citations from biomedical literature, including journals indexed in MEDLINE. Researchers trust the information in this database [20,23], highlighting its central role in accessing medical literature. However, it should be noted that no single database contains access to all records in a given field-there is no universal database [24,25]. Therefore, according to many guidelines (e.g., for secondary research), it is recommended to use multiple sources to gain a comprehensive view of the issue [25,26].

Nevertheless, limited familiarity with tools like the PICO framework indicates substantial gaps in knowledge that hinder the effective utilisation of these resources; however, increasing the salary level could reduce this problem. Some studies show that many participants (including pharmacists) struggle to effectively use the PICO framework for formulating clinical questions and finding relevant scientific articles [27], and one of the challenges for pharmacists is the lack of proper training in this area [28].

Participants in the study were also asked to rank the importance of different sections of a scientific publication. Respondents placed the most importance on the title and abstract (Fig 1), which may indicate a tendency to base the evaluation of a publication on freely available elements (e.g., in a PubMed record). On the other hand, analysing the title and abstract is a natural first step in assessing a scientific publication, such as conducting a systematic review. Research published in BioMed Central (BMC) shows that most people rely on primary search engines, which often contain only the title and abstract of articles, highlighting their importance in the evaluation process [29]. Conversely, respondents considered study limitations, the bibliography, and the discussion section the least important.

Interestingly, the methodology was not deemed one of the most essential elements of a publication, and only half of the respondents paid greater attention to it (as in the previous question). This criterion should be ranked much higher in the hierarchy of importance. The authors of the publication 'A Tutorial on Methodological Studies' [20] particularly emphasise how important it is to plan and describe the methodology precisely to ensure the high quality and credibility of the research. Similarly, the issue of study limitations-while respondents regarded them as almost marginal in importance-is a critical factor often overlooked. Study limitations highlight potential weaknesses and areas that may render the presented results less reliable. Identifying and discussing limitations also increases transparency and credibility, which is important for the further development of scientific research. Highlighting limitations is also crucial regarding research transparency and ethics [30]. In the context of the obtained results, it is also notable that respondents ranked the introduction of a publication as more important than the discussion, even though this section is often considered one of the most critical parts of a scientific article, as it answers how the study results contribute to knowledge development in a given field [30].

Regarding the hierarchy of scientific evidence, respondents ranked systematic reviews, meta-analyses, and randomised controlled trials as studies with the highest reliability, followed by cohort studies and case-control studies as moderately reliable, and case series and case reports as low-reliability studies (Fig 2), which aligns with EBPharm/EBM principles [31,32].

## Limitations

The study's main limitation was that most of the obtained results came from online distribution (using social media) rather than from direct distribution. Additionally, in-person surveys were delivered to a limited number of pharmacies in only two large cities and one medium-sized city. Furthermore, the fact that the survey was available only in an electronic version undoubtedly affected the age structure of the respondents (individuals up to 40 years old constituted over 80%). It is worth noting that parameters such as the respondents' gender (over 82% females) are not a limitation but rather reflect the actual employment situation in this profession in Poland. Furthermore, most responses were obtained from pharmacists and pharmacy technicians working in community pharmacies, and almost half of the respondents had less than five years of work experience. The study group should be expanded to include individuals working in other positions (e.g., hospital pharmacists) and those with more extended work experience to gain a more comprehensive insight into Evidence-Based Pharmacy. Moreover, in the subquestions (like 12.1, 20.1, 20.2, 21.1), the number of responses was generally lower than in the main question, which was directly due to the responses provided by the participants in the main questions. Additionally, some questions were intentionally designed to be very general, e.g., "Do you apply the principles of Evidence-Based Pharmacy (based on the concept of Evidence-Based Medicine) in your professional practice?", which was meant to verify whether respondents apply these principles in any way and whether they are familiar with this terminology at all. This study used an exploratory approach based primarily on Pearson's Chi² tests. While this method is appropriate for identifying associations in cross-sectional data, two additional limitations must be acknowledged. First, the lack of multivariable analyses (e.g., logistic regression) limits the ability to control for potential confounding factors such as age, education level, or work experience. Second, no correction for multiple comparisons was applied, which increases the risk of Type I errors (false positives). However, strict corrections in an exploratory context could substantially increase the risk of Type II errors (false negatives). Therefore, all reported associations should be interpreted with caution. Future analyses will consider both multivariable models and statistical corrections to enhance the robustness of findings

## Implications and recommendations

1. Enhanced Training Programs: Developing robust training modules focusing on systematic reviews, advanced evidence evaluation techniques, and effective use of databases is critical.

2. Integrating EBPharm into Continuous Professional Development (CPD): Regular workshops and courses tailored to experienced pharmacists can address knowledge attrition and maintain competence.

3. Promotion of Reliable Information Sources: Initiatives should be prioritised to reduce reliance on informal sources, especially among pharmacy technicians. Ensuring access to key databases like PubMed, Cochrane Library, and Scopus can further support evidence-based practices.

4. Financial Support for Professional Development: Policies incentivising financial investment in education, such as subsidies for advanced training programs, may enhance knowledge acquisition and skill development.

5. Interdisciplinary Collaboration: Encouraging collaboration between academia and professional organisations can facilitate the dissemination of best practices and emerging evidence.

## Conclusions

In conclusion, although most respondents have a positive attitude towards Evidence-Based Pharmacy (EBPharm) and claim to apply these principles in practice, the survey results reveal significant gaps between perceived and actual knowledge and skills in EBPharm. The findings also suggest a need for better-tailored and expanded educational programmes and training to more effectively prepare this professional group to apply EBPharm principles in their daily practice,

emphasising continuous evaluation and development of their competencies. Future efforts in this field should focus on acquiring theoretical and practical knowledge of available resources and information sources, enabling pharmacists to effectively and efficiently apply EBPharm principles in their daily work.

## Supporting information

**S1 Table. Statistical analysis – additional results (correlations between questions).**
(DOCX)

**S1 File. Survey.** Survey questionnaire.
(DOCX)

**S2 File. Survey data.** Raw data.
(XLSX)

## Author contributions

**Conceptualization:** Piotr Ratajczak, Krzysztof Kus.

**Data curation:** Piotr Ratajczak, Barbara Jacieczko, Bartosz Kędziora, Dorota Kopciuch, Anna Paczkowska, Tomasz Zaprutko.

**Formal analysis:** Piotr Ratajczak, Barbara Jacieczko.

**Investigation:** Piotr Ratajczak, Barbara Jacieczko.

**Methodology:** Piotr Ratajczak, Krzysztof Kus.

**Resources:** Piotr Ratajczak, Bartosz Kędziora, Krzysztof Kus.

**Software:** Piotr Ratajczak.

**Supervision:** Piotr Ratajczak, Krzysztof Kus.

**Validation:** Piotr Ratajczak, Barbara Jacieczko, Bartosz Kędziora, Dorota Kopciuch, Anna Paczkowska, Tomasz Zaprutko, Krzysztof Kus.

**Visualization:** Piotr Ratajczak, Barbara Jacieczko, Bartosz Kędziora.

**Writing – original draft:** Piotr Ratajczak, Barbara Jacieczko, Bartosz Kędziora, Dorota Kopciuch, Anna Paczkowska, Tomasz Zaprutko, Krzysztof Kus.

**Writing – review & editing:** Piotr Ratajczak, Bartosz Kędziora, Dorota Kopciuch, Anna Paczkowska, Tomasz Zaprutko, Krzysztof Kus.

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
