## [Decision Letter · Decision Letter 0]

Dear Dr. Ratajczak,

Thank you for submitting your manuscript to PLOS ONE. After careful consideration, we feel that it has merit but does not fully meet PLOS ONE’s publication criteria as it currently stands. Therefore, we invite you to submit a revised version of the manuscript that addresses the points raised during the review process.

We look forward to receiving your revised manuscript.

Kind regards,

Ebenezer Wiafe, PhD, MPharm, Pharm D

Academic Editor

PLOS ONE

Additional Editor Comments:

Please, pay critical attention to Reviewer 1.

Reviewers' comments:

Reviewer's Responses to Questions

**Comments to the Author**

1. Is the manuscript technically sound, and do the data support the conclusions?

Reviewer #1: Yes

Reviewer #2: Yes

2. Has the statistical analysis been performed appropriately and rigorously?

Reviewer #1: No

Reviewer #2: Yes

3. Have the authors made all data underlying the findings in their manuscript fully available?

Reviewer #1: Yes

Reviewer #2: Yes

4. Is the manuscript presented in an intelligible fashion and written in standard English?

Reviewer #1: Yes

Reviewer #2: Yes

Reviewer #1: This manuscript explores a topic: how Evidence-Based Pharmacy (EBPharm) influences pharmaceutical care quality. The topic is timely and important, however, there are several issues that need to be addressed to strengthen the scientific rigor and transparency of the study. The following specific comments are provided for the authors’ consideration:

1. Please elaborate on the process of questionnaire construction. How were the items selected? Why did you decide on 26 primary items (why not more or fewer)? Furthermore, the response options across questions vary (e.g., two, three, or five choices). What was the rationale behind the choice of response scales for each item?

2. There is no mention of whether the questionnaire underwent any psychometric validation. Was the instrument tested for validity (Ex: content, face, construct validity) and reliability (internal consistency)? Please provide this information in the method section.

3. While the manuscript refers to items related toEBPharm, scientific information sources, and assessment of publication quality, it does not specify how many questions were assigned to each domain. Please provide a clear mapping of the 26 questions to their respective conceptual domains.

4. Several multiple-choice items were reportedly converted to binary (yes/no) variables. Please explain how this recoding was done and justify the decision.

5. What is the distinction between a “pharmacist in an open pharmacy” and a “pharmacist in a closed pharmacy”? This terminology may not be familiar to international readers and should be explained.

6. What does "pharmacy specialisation" refer to in this context? Is it a formal postgraduate qualification, a board certification, or something else?

7. Given the use of social media as a recruitment method, how did you verify that only eligible participants (pharmacists or pharmacy technicians) completed the survey? Is there a possibility that non-pharmacists participated?

8. There is no information about how many individuals were invited to participate, nor the response rate. This is critical for assessing the representativeness of the sample.

9. The eligibility criteria for participation should be explicitly stated in the Methods section.

10. Were there any missing responses in the dataset? If so, how were they handled?

11. Please clarify which geographic or professional areas were targeted for recruitment. Were specific regions or pharmacy sectors prioritized?

12. All statistical analyses rely on Pearson’s Chi² test, with no multivariate or adjusted regression models used. This limits the ability to control for potential confounders such as age, education level, or income. For example, the observed relationship between salary and EBPharm knowledge may be confounded by years of experience or educational attainment. Consider using logistic regression or similar techniques to adjust for such factors.

Reviewer #2: This manuscript addresses a relevant and timely topic: the integration and impact of evidence-based pharmacy (EBPharm) practices among Polish pharmacists. The authors present valuable insights from a well-structured survey and provide a comprehensive discussion on the implications of their findings.

However, while the paper is informative and rich in data, some areas require further clarification, refinement in writing, and minor structural enhancements to meet the standards of publication in PLOS ONE.

Recommendation

Minor Revisions Required

**Do you want your identity to be public for this peer review?** For information about this choice, including consent withdrawal, please see our Privacy Policy

Reviewer #1: No

Reviewer #2: **Yes: ** Amal K Suleiman

---

## [Author Response · Author response to Decision Letter 1]

30 May 2025

Reviewer 1

We sincerely thank the Reviewer for thorough analysis and valuable comments. Our response is structured according to the points raised in the review.

1. Please elaborate on the process of questionnaire construction. How were the items selected? Why did you decide on 26 primary items (why not more or fewer)? Furthermore, the response options across questions vary (e.g., two, three, or five choices). What was the rationale behind the choice of response scales for each item?

Thank you for highlighting this important aspect. We have also included this response in the manuscript (Study Design).

The questionnaire was developed based on a literature review in EBPharm, pharmacy education, and information literacy. Three key domains were identified: (1) knowledge and attitudes toward EBPharm, (2) use of scientific sources, and (3) the ability to evaluate the quality of scientific publications.

In case of response options:

The variation in response scales was intentional and based on each question's nature and cognitive demand. Factual and eligibility-related questions (e.g., use of EBPharm, familiarity with the PICO framework) were assigned binary (yes/no) scales to ensure clarity and avoid unnecessary ambiguity.

Questions that required self-assessment of knowledge or competencies (e.g., self-rated EBPharm knowledge, ability to assess scientific evidence) employed Likert-type scales, which allow respondents to express varying degrees of confidence or perceived proficiency. These scales help capture the intensity or nuance of the respondent’s experience rather than the simple presence or absence of a trait.

In contrast, for questions related to the frequency of behaviour (e.g., updating knowledge, using databases), three-point ordinal scales (e.g., always / sometimes / never) were used to balance granularity with respondent burden. A three-point scale was selected instead of a more detailed five-point frequency scale to encourage completion and reduce response fatigue, especially given the short expected survey duration (5–10 minutes).

Each scale type was chosen to optimise response validity, match the measurement objective of the item, and maintain survey usability.

This statement was introduced in the Data Collection and Research Method sections:

Answer scales were adapted to question types: binary for factual items, Likert-type for self-assessment, and three-point scales for frequency-based questions.

2. There is no mention of whether the questionnaire underwent any psychometric validation. Was the instrument tested for validity (Ex: content, face, construct validity) and reliability (internal consistency)? Please provide this information in the method section.

Thank you for your valuable comment. This aspect has also been addressed (in the Study Design section) and supplemented with statistical details, in line with the reviewer's suggestion.

The creation of the tool was based on expert consultation and pilot testing to ensure thematic balance and respondent feasibility. To address content and face validity, all items were reviewed by academic experts in pharmacy and education. The questionnaire was internally tested within our academic unit before distribution.

3. While the manuscript refers to items related toEBPharm, scientific information sources, and assessment of publication quality, it does not specify how many questions were assigned to each domain. Please provide a clear mapping of the 26 questions to their respective conceptual domains.

Thank you for this comment – below we present the allocation of questions to the three main thematic domains:

• knowledge and attitudes toward EBPharm (questions 10–14),

• use of scientific sources (questions 15–22),

• the ability to evaluate the quality of scientific publications (questions 23–25).

This information has also been incorporated into the manuscript – in the Data Collection and Research Method section.

4. Several multiple-choice items were reportedly converted to binary (yes/no) variables. Please explain how this recoding was done and justify the decision.

In the case of multiple-choice questions, each response option was coded as a separate binary variable (1 = selected, 0 = not selected). This approach allowed for more precise statistical analysis of associations between specific answers and other variables (e.g., age, education level) using Pearson’s Chi² test. The method follows best practices in survey data analysis and is described in the "Statistical Analysis" section of the manuscript.

For example, question 21.1 asked respondents which medical databases they use. Each selected database (e.g., PubMed, Embase, Cochrane Library) was treated as a separate binary variable. These were then used to compare usage patterns across subgroups such as income or EBPharm training status. Similarly, other multiple-choice questions like 5, 17 or 20.2.

This binary coding enabled us to isolate and assess the significance of individual selections rather than aggregating diverse options into a single measure, thereby increasing the clarity and granularity of the analysis. All binary variables and their descriptive labels are available in the supplementary dataset (Survey raw data.xlsx).

5. What is the distinction between a “pharmacist in an open pharmacy” and a “pharmacist in a closed pharmacy”? This terminology may not be familiar to international readers and should be explained.

“Open pharmacy” refers to publicly accessible community pharmacies, while “closed pharmacy” denotes pharmacies operating within restricted institutions such as hospitals. This terminology has been clarified in the Data collection and research method section.

6. What does "pharmacy specialisation" refer to in this context? Is it a formal postgraduate qualification, a board certification, or something else?

In this study, the term “pharmacy specialisation” refers to formal postgraduate specialisation training in Poland (e.g., in hospital pharmacy, clinical pharmacy, pharmacology, pharmacoeconomics), conducted following regulations issued by the Ministry of Health and concluded with a state specialisation examination. This information was provided in the Data collection and research method section.

7. Given the use of social media as a recruitment method, how did you verify that only eligible participants (pharmacists or pharmacy technicians) completed the survey? Is there a possibility that non-pharmacists participated?

To verify respondent eligibility, screening questions regarding education level, job position, professional experience, and salary were included. Additionally, the introduction to the survey clearly stated that it was intended exclusively for individuals working in pharmacies and pharmacy students currently undergoing professional internships. The survey was distributed within closed professional groups on social media and in pharmacies (by sharing the survey link). This limitation has been acknowledged in the Limitations section.

8. There is no information about how many individuals were invited to participate, nor the response rate. This is critical for assessing the representativeness of the sample.

Due to the partially open recruitment method (via social media), it was not possible to estimate the number of individuals invited to participate in the study, and therefore the exact response rate could not be determined. However, a minimum sample size calculation was performed and presented in the manuscript to ensure adequate statistical power. The required minimum of 266 responses was calculated based on data from Statistics Poland (2024), indicating a total of 57,800 individuals employed in the relevant sector (this information is provided in the statistical analysis section). This limitation has been acknowledged in the Limitations section.

9. The eligibility criteria for participation should be explicitly stated in the Methods section.

The inclusion criteria for the study were: working as a pharmacist or pharmacy technician in Poland, or being a pharmacy student currently undergoing a professional internship. This information has been added to the Data collection and research method section.

10. Were there any missing responses in the dataset? If so, how were they handled?

There were no missing data in the dataset. All empty cell values in the "Supplement Survey data" file were related solely to questions that were logically dependent on previous answers. For example, the question about specialisation appeared only to respondents who had declared they possessed one. Similarly, questions regarding specific databases were shown only to those who reported using such resources. The appropriate sample size (n) was applied for each variable in all analyses accordingly.

11. Please clarify which geographic or professional areas were targeted for recruitment. Were specific regions or pharmacy sectors prioritised?

The recruitment primarily included pharmacists and pharmacy technicians from community pharmacies. Paper questionnaires were distributed directly in pharmacies located in three cities (two large and one medium-sized), while the majority of responses were obtained through online recruitment. No regional or sectoral priorities were applied. This information is provided in the Limitations section.

12. All statistical analyses rely on Pearson’s Chi² test, with no multivariate or adjusted regression models used. This limits the ability to control for potential confounders such as age, education level, or income. For example, the observed relationship between salary and EBPharm knowledge may be confounded by years of experience or educational attainment. Consider using logistic regression or similar techniques to adjust for such factors.

Thank you for the suggestion. We agree that multivariable analysis would allow for better control of potential confounding variables. The present study applied an exploratory approach based on Pearson’s Chi² tests, given the study's scope and the variables' categorical nature. We acknowledge that logistic regression or other multivariate models may yield more in-depth insights and are being considered for the next stages of data analysis. This limitation has been noted in the updated Limitations section.

We want to emphasise that additionally effect sizes using Cramer V and 95% confidence intervals were calculated to enhance the interpretability of the Chi² results.

Reviewer 2

We sincerely thank the Reviewer for thorough analysis and valuable comments. Our response is structured according to the points raised in the review.

1. While the manuscript claims high adoption of EBPharm (e.g., 98.21% report applying it), this seems to contradict the significant knowledge gaps (e.g., only 25.88% familiar with PICO; 3.83% able to conduct systematic reviews). These findings need more nuanced interpretation—possibly indicating a discrepancy between perceived and actual application.

Suggest adding a paragraph discussing the self-reporting bias and the possible overestimation of EBPharm usage.

Thank you for highlighting this important aspect. A relevant passage addressing both of the reviewer's comments has been incorporated into the Discussion section.

This inconsistency in the results becomes even more apparent when we consider that only a limited proportion of participants declared familiarity with the PICO framework, a fundamental tool used to formulate clinical questions and evaluate evidence. Knowledge of this framework is essential in the field of EBPharm, and such a contrast highlights a potential discrepancy between the perceived and actual implementation of expertise in this area. Although respondents may identify with the EBPharm concept, their understanding often appears superficial, based more on general awareness of the terminology than on genuine training. It may indicate the presence of self-reporting bias, where factors such as professional expectations lead individuals to overestimate their competencies. It points to the need to expand professional training for pharmacists.

2. Survey Instrument

The development and validation process of the survey is not clearly described. Was it pilot-tested? Were reliability or validity metrics (e.g., Cronbach’s alpha) assessed?

Please include this detail in the Materials and Methods section.

Thank you for your valuable comment. This aspect has also been addressed in the Study Design section in line with the reviewer's suggestion.

The creation of the tool was based on expert consultation and pilot testing to ensure thematic balance and respondent feasibility. To address content and face validity, all items were reviewed by academic experts in pharmacy and education. The questionnaire was internally tested within our academic unit before distribution.

Additionally, the internal consistency of the 7-item publication credibility scale (questions 24a–24g) was assessed using Cronbach’s alpha. This was the only section eligible for such analysis, as the items shared a unified Likert-type ordinal scale and a consistent conceptual focus—namely, the perceived reliability of different types of scientific studies. The obtained coefficient was α = 0.611, indicating marginal internal consistency. This result likely reflects genuine differences in how respondents evaluate the credibility of various study designs, rather than measurement error. Other questionnaire items were unsuitable for Cronbach’s alpha calculation due to their heterogeneous formats (binary, factual, ordinal) and independent, single-item structure. As the analysis applies only to one specific multi-item matrix, we decided not to report this result in the main body of the manuscript but include it in the reviewer response instead.

3. Statistical Analysis Details

The text describes Pearson’s Chi-square results, but a table summarizing effect sizes and confidence intervals would improve interpretability.

Based on the Reviewer's comments, we have included those data in the manuscript (Table 4) and supplementary file (S1 Table).

Also, the authors should clarify how multiple testing was handled—was any adjustment made (e.g., Bonferroni correction)

Thank you for this comment. As the study was exploratory in nature and aimed to identify potential associations rather than test predefined hypotheses, no adjustment for multiple testing (such as Bonferroni correction) was applied. Due to the exploratory nature of the analysis, results were interpreted at the conventional threshold of p < 0.05 without correction. Applying strict correction methods in this context could have increased the risk of Type II errors (false negatives), potentially obscuring meaningful patterns. This limitation has been acknowledged in the updated Limitations section of the manuscript.

4. The manuscript is generally well-written but would benefit from a light language polish, particularly to improve flow and reduce redundancy in the Discussion section. Example: Avoid repeating the same statistical results in both the Results and Discussion without added interpretation

Thank you for this helpful suggestion. We have revised the Discussion section to improve its clarity and reduce redundancy. Specifically, we removed repeated statistical values (e.g., Chi² and p-values) that were already presented in the Results section and focused instead on interpreting the significance and implications of the findings.

Editor

The manuscript has been revised to comply with PLOS ONE’s formatting and file naming

guidelines, following the style templates provided.

Captions for

---

## [Decision Letter · Decision Letter 1]

The Impact of Evidence-Based Pharmacy on the Quality of Pharmaceutical Care. A Survey Study

PONE-D-25-12319R1

Dear Dr. Ratajczak,

We’re pleased to inform you that your manuscript has been judged scientifically suitable for publication and will be formally accepted for publication once it meets all outstanding technical requirements.

Kind regards,

Ebenezer Wiafe, PhD, MPharm, Pharm D

Academic Editor

PLOS ONE

Additional Editor Comments (optional):

Please, pay attention to how you reported your *p* -values. 

I am happy to have served you.

Reviewers' comments:

Reviewer's Responses to Questions

**Comments to the Author**

Reviewer #1: All comments have been addressed

Reviewer #2: All comments have been addressed

2. Is the manuscript technically sound, and do the data support the conclusions?

Reviewer #1: Yes

Reviewer #2: Yes

3. Has the statistical analysis been performed appropriately and rigorously?

Reviewer #1: Yes

Reviewer #2: Yes

4. Have the authors made all data underlying the findings in their manuscript fully available?

Reviewer #1: Yes

Reviewer #2: Yes

5. Is the manuscript presented in an intelligible fashion and written in standard English?

Reviewer #1: Yes

Reviewer #2: Yes

Reviewer #1: Only a minor suggestion: do not write p = 0.0000, as it implies p = 0, which is not the case. You should write p < 0.001 instead.

Reviewer #2: authors have adequately addressed the comments raised in a previous round of review and now can accept the paper

**Do you want your identity to be public for this peer review?** For information about this choice, including consent withdrawal, please see our Privacy Policy

Reviewer #1: No

Reviewer #2: **Yes: ** Amal K Suleiman

---

## [Editor Report · Acceptance letter]

PONE-D-25-12319R1

PLOS ONE

Dear Dr. Ratajczak,

I'm pleased to inform you that your manuscript has been deemed suitable for publication in PLOS ONE. Congratulations! Your manuscript is now being handed over to our production team.

Kind regards,

on behalf of

Dr. Ebenezer Wiafe

Academic Editor

PLOS ONE